# Bearing Fault Diagnosis Based on Measured Data Online Processing, Domain Fusion, and ANFIS

Quang Thinh Tran [1] and Sy Dzung Nguyen [2,3,*]

1   Faculty of Mechanical Engineering, Industrial University of Ho Chi Minh City,
    Ho Chi Minh City 700000, Vietnam
2   Faculty of Electrical and Electronics Engineering, Ton Duc Thang University,
    Ho Chi Minh City 700000, Vietnam
3   Division of Computational Mechatronics, Institute for Computational Science, Ton Duc Thang University,
    Ho Chi Minh City 700000, Vietnam
*   Correspondence: nguyensydung@tdtu.edu.vn

**Abstract:** Processing noise online in sensors-based measurement data (SMD) and mitigating the effect of domain drift are always challenges. As a result, it negatively impacts the effectiveness and feasibility of data-driven model (DDM)-based mechanical-system fault identification (MFI). Here, we propose an online bearing fault diagnosis method named ANFIS-BFDM by using an adaptive neurofuzzy inference system (ANFIS). Reduction in the influence of domain drift between the source domain and target domain (DDSTD) is considered in both the data processing and fault identification. Online solutions for preprocessing SMD and exploiting the filtered data to label the target domain are presented in a fusion domain deriving from the source and target domains. First, in the offline phase, frequency-based splitting of SMD into different time series is performed to cancel the high-frequency region. An optimal data screening threshold (ODST) is distilled in the remaining low-frequency data to develop an impulse noise filter named FIN. An ANFIS then identifies the dynamic response of the bearing(s) via the filtered data. The FIN and ANFIS are finally exploited during the online phase to filter noise and recognize the object's health status online. The survey results reflect the positive effects of the method, even if severe impulse noise appears in the databases.

**Keywords:** bearing fault diagnosis; deep-learning-based fault identification; AI-based fault diagnosis

## 1. Introduction

The vibration signal contains meaningful information about the appearance of fault/damage on mechanical systems [1,2]. Therefore, it has been widely used for DDM-based MFI [3–6]. Accordingly, a data matrix $\mathbf{X}^s$ labeled via vector $\mathbf{y^s}$ (called source domain) is built in the offline phase to identify the dynamic response of the managed mechanical object (MMO), whereas a data matrix $\mathbf{X}^t$ (called target domain) is set up online to reflect the dynamic behavior of the MMO at the survey time. With advantages in managing inaccurate and uncertain databases, fuzzy logic (FL) and artificial neural networks (ANN) are employed extensively for this approach [7–10]. Reality has shown that a fit fuzzy-set space for FL and a suitable net structure for ANN need to be pre-estlablished in each application. However, a satisfactory response to these requirements is often a considerable challenge [11]. Additionally, processing noise online in SMD and mitigating the negative effect of domain drift between $\mathbf{X}^s$ and $\mathbf{X}^t$ are always difficult [12,13].

ANFIS is a fuzzy model that is set up automatically through the ANN's training process. Due to being able to partly subdue the inherent difficulties of FL and ANN, it has been widely used, including the AI-based fault diagnosis [7,10]. ANFIS can also collaborate with singular spectrum analysis (SSA) successfully in applications related to time series [11]. In such an approach, filtering noise in measured data or seeking information in data series are typical applications [14–16]. For example, a continuous hidden Markov model to extract

the bearing fault features relies on the singular features via SSA [17], or processing data for an ANFIS-based classification derives from SSA [11].

Noise in measurement data related to the measurement method and instrument errors, false observations, changing environmental conditions, or random aspects cause random IN (impulse noise). There is, however, still no sufficiently strong solution for canceling IN in measured data [18]. This difficulty is because IN occurs randomly related to a group of factors whose circumstances and the mechanism that affects measurement accuracy are unknown; the source of these errors is unclear and not easy to find. Nguyen et al. [19] presented a method of filtering random noise and IN in data derived from the dynamic response of smart dampers such as magnetorheological and electrorheological dampers. The filter relied on an optimal data screening threshold quantified via clustering results. Unfortunately, its narrow application scope is only the databases derived from the dynamic response of smart dampers.

Along with the negative impact of noise, DDSTD in the databases for MFI always exists, which reduces the effectiveness and feasibility of DDM-based MFIs in practical applications. In many cases, DDSTD is the main cause for a method with appropriate accuracy in the theoretical investigation, but not in its applications with practical operating conditions. It is the strong motivation attracting researchers to pay attention to seeking solutions for domain adaptation (DA) to take part in compensating adaptively for domain drift [3–6,13]. Li et al. [3] proposed a bearing fault diagnosis based on ANN and cross-domain. Wu et al. [4] showed a method of bearing fault diagnosis that trained an ANN from the cross-domain to reduce the distribution difference between the source domain and target domain in each data channel. Another DA, called transfer Component analysis (Pal et al. [13]), was also applied effectively for pattern analysis. It learned transfer components across domains in a reproducing kernel Hilbert space. In the subspace spanned by these transfer components, data properties are preserved, and data distributions in different domains become closer to each other. However, either generating the fake samples as shown by Li et al. [3] or adjusting the ANN weights by Wu et al. [4] leads to a disadvantage related to the cumulated error from the network-based MFIs. It is exacerbated under the influence of noise. Additionally, the impact of IN is the considerable difficulty encountered in the method of transfer component analysis, as shown by Pan et al. [13].

Consequently, we propose an online bearing fault diagnosis method relying on ANFIS, named ANFIS-BFDM in this research, by seeking fit solutions for the aspects noted above. Namely, we (i) set up a fusion domain deriving from the source and target domains, and (ii) carry out tasks of processing measured data and exploiting the filtered data to classify/label $_u\boldsymbol{X}^t$ via solutions understood as performing embedding processes in the fusion domain, $\boldsymbol{X}^f = \boldsymbol{X}^s \cup {}_l\boldsymbol{X}^t$, where $_l\boldsymbol{X}^t \subset \boldsymbol{X}^t$ is the data matrix containing the labeled samples (in the label vector $_l\boldsymbol{y}^t$) corresponding to the bearing's healthy condition, and the remainder in $\boldsymbol{X}^t$ denoted $_u\boldsymbol{X}^t$ is the unlabeled samples. These works can partially diminish the influence of DDSTD in data processing and fault identification. The ANFIS-BFDM has offline and online phases. First, in the offline, frequency-based splitting of SMDs into different time series is performed to cancel the high-frequency region. An optimal data screening threshold ODST is then distilled in the remaining low-frequency data, to which we develop an impulse noise filter named FIN. From SMDs processed by the FIN, the dynamic response of the managed bearing(s) is identified via an ANFIS. Finally, we exploit the FIN and the trained ANFIS in the online phase to filter noise and recognize the object's health status online.

The three main contributions of this study follow. The first is the FIN. It follows the idea shown by Nguyen et al. [19], where the change in the distribution of the cluster data space built from the measured data stream is employed to catch IN. However, instead of focusing on the source domain only as in this reference, we seek the change in the embedding data space called the fusion domain. It relies on an observation that the fusion domain can track the measured data stream better than the source domain. The reason is the existing domain drift between the source and target domains.

The second contribution is a solution of combination between SSA and the filter FIN to extend the frequency range of processed data. This aspect is a vital supplementation for the previous research [11], where the high-frequency noise is filtered only.

The third contribution is the algorithm ANFIS-BFDM for identifying damage of the bearing(s). It consists in processing SMD streams based on SSA and the FIN, establishing a multifeature from the processed data, and the ANFIS-based interpolation (1), where the ANFIS takes the role of a mapping from the fusion domain $\mathbf{X}^f$ to the target domain $\mathbf{X}^t$:

$$\begin{aligned} \text{ANFIS}: \quad & \mathbf{X}^f \to \mathbf{X}^t \\ & {}_u\mathbf{X}^t \mapsto \hat{\mathbf{y}} = \text{ANFIS}\left({}_u\mathbf{X}^t\right), \end{aligned} \tag{1}$$

where $\hat{\mathbf{y}}$ is the output vector of the ANFIS corresponding to the input ${}_u\mathbf{X}^t$. In this relationship, instead of using a data-driven model to identify the source domain $\mathbf{X}^s$, as shown by Lei et al. [10] and Tran et al. [11], we exploit $\mathbf{X}^f$ to make the negative impacting degree of domain drift between the source and target domains weaken.

During this paper, we use the abbreviations and symbols in Table 1.

**Table 1.** Abbreviations and symbols.

| Abbreviation | Full Phrase | Symbol | Meaning |
|---|---|---|---|
| ANFIS | Adaptive neurofuzzy inference system | $d_{ak}$ | Convergent degree |
| ANFIS-BFDM | BFDM based on ANFIS | $d_k$ | Dispersion of data points in the $k$-th cluster |
| BFDM | Bearing fault diagnosis method | $L$ | Length of time series transformed by the SSA |
| CDS | Cluster data space | H | Number of considered fault statuses |
| DA | Domain adaptation | $p$ | Number of time series transformed by the SSA |
| DDM | Data-driven model | $q$ | Number of low frequency series deriving from SSA |
| DDSTD | Domain drift between the source domain and target domain | $\widetilde{P}_0$ | Number of the established samples corresponding to only one health's single status of the managed object |
| FIN | Impulse noise filter | $R_k$ | Normalized distribution radius |
| MFI | Mechanical system fault identification | $R_{A^{(k)}}$ | Distribution radius of $k$-th data cluster |
| ODST | Optimal data screening threshold | $\mathbf{X}^s$ | Data matrix in the labeled source domain |
| SMD | Sensors-based measurement data | $\mathbf{X}^t$ | Data matrix in the target domain |
| SSA | Singular spectrum analysis | ${}_l\mathbf{X}^t$ | Labeled samples in $\mathbf{X}^t$ |
| IDS | Initial data set | ${}_u\mathbf{X}^t$ | Unlabeled samples in $\mathbf{X}^t$ |
| IN | Impulse noise | $\mathbf{y^s}$ | Label of the source domain |
| | | ${}_l\mathbf{y}^t$ | Label of ${}_l\mathbf{X}^t$ |
| | | $\Gamma^k$ | The $k$-th data cluster |

## 2. Related Works

Let us consider a given initial data set (IDS), IDS : $\left(\mathbf{X}(\bar{\mathbf{x}}_i) \in \Re^{P \times n}, \mathbf{y}(y_i) \in \Re^{P \times 1}\right)$ consisting of input–output data points $\left(\bar{\mathbf{x}}_i, y_i\right), i = 1 \ldots P$, where $\bar{\mathbf{x}}_i = [x_{i1}, \ldots, x_{in}] \in \Re^n$ belongs to the input data space $\mathbf{X}$ and $y_i \in \Re^1$ belongs to the output data space $\mathbf{y}$. The IDS is a measured database with noise, including random and impulse noise (IN), expressing an unknown mapping $f: \text{X} \to \text{Y}$.

This section shows (i) an approximation of the mapping $f$ using ANFIS, and (ii) the method for filtering IN based on ANFIS. These contents are exploited in the proposed theory, described in Section 3.

### 2.1. Building ANFIS from a Database

The algorithm named ANFIS-JS (Nguyen et al. [16]) for building an ANFIS from the joined input–output data space IDS is presented here. It is exploited in Section 3 to (i) set up a cluster data space (CDS) for our proposed filter (named FIN) for filtering IN in mechanical vibration databases, and (ii) build ANFISs for our proposed method of DDM-based fault diagnosis. The algorithm for kernel fuzzy *C*-means clustering with kernelization (KFCM-K) (Marcelo et al. [20]) is adopted here. Accordingly, clusters $\Gamma^k, k = 1 \ldots C$, are established

through seeking cluster centers $\overset{-0}{\mathbf{x}}_1, \ldots, \overset{-0}{\mathbf{x}}_C$ ($\overset{-0}{\mathbf{x}}_i = [x^0{}_{i1}, \ldots, x^0{}_{in}] \in \Re^n$) in the input space such that $J_{KFCM}(.) \rightarrow \min$, where

$$
\begin{cases}
J_{KFCM}\left(\mu_{ij}, \overset{-0}{\mathbf{x}}\right) = \sum_{i=1}^{C} \sum_{j=1}^{P} \mu_{ij}{}^m \left\| \phi(\overset{-}{\mathbf{x}}_j) - \phi(\overset{-0}{\mathbf{x}}_i) \right\|^2 \\
\text{subjected to } \left( \mu_{ij} \in [0,1] \; \forall i, \; j \; ; \; \sum_{i=1}^{C} \mu_{ij} = 1 \; \forall j \right),
\end{cases}
\tag{2}
$$

$\phi(.)$ is the kernel function and $m > 1$ is the fuzzy factor. In the case of using Gaussian kernel function $K(.)$, the update law of the cluster centroids $\overset{-0}{\mathbf{x}}_1, \ldots, \overset{-0}{\mathbf{x}}_C$ and the membership degree of the *j*-th data point for the *i*-th cluster ($\mu_{ij}$) can be inferred from (2) as follows:

$$
\overset{-0}{\mathbf{x}}_i = \frac{\sum_{j=1}^{P} \mu_{ij}{}^m \; \overset{-}{\mathbf{x}}_j \; K(\overset{-}{\mathbf{x}}_j, \overset{-0}{\mathbf{x}}_i)}{\sum_{j=1}^{P} \mu_{ij}{}^m K(\overset{-}{\mathbf{x}}_j, \overset{-0}{\mathbf{x}}_i)},
\tag{3}
$$

$$
\mu_{ij} = \begin{cases}
\left[ \left( \sum_{h=1}^{C} \frac{1 - K(\overset{-}{\mathbf{x}}_j, \overset{-0}{\mathbf{x}}_i)}{1 - K(\overset{-}{\mathbf{x}}_j, \overset{-0}{\mathbf{x}}_h)} \right)^{1/(m-1)} \right]^{-1} & \text{if } \overset{-}{\mathbf{x}}_j \neq \overset{-0}{\mathbf{x}}_i \\
1 \text{ (and } \mu_{ik(k \neq j)} = 0) \text{ if } \overset{-}{\mathbf{x}}_j = \overset{-0}{\mathbf{x}}_i \\
(i = 1 \ldots C; j = 1 \ldots P).
\end{cases}
\tag{4}
$$

The hard distribution in the CDS can be set up. The *j*-th data point is called to be distributed hardly into the *q*-th cluster if $\mu_{qj} = \max\limits_{h=1 \ldots C} \left( \mu_{hj} \right)$. In this hard distribution, the created data clusters are the so-called hard clusters.

*2.2. Optimal Data Screening Threshold*

This subsection summarizes the algorithm AfODST (Nguyen et al. [19]) called Algorithm 1 in this paper. The algorithm takes part in the proposed new filter named FIN in the following section.

2.2.1. Related Definitions

First, the given IDS is re-formed as a new database: IDS : $\left( \mathbf{R_X} \in \Re^{P \times (n+1)}, \mathbf{y} \in \Re^{P \times 1} \right)$, in which

$$
\mathbf{R_X} = \left\{ \overset{-}{\mathbf{x}}_i \; \widetilde{y}_i = \frac{y_i}{\max|y_i|} \right\} = \{ x_{i1} \; x_{i2} \ldots x_{in} \; \widetilde{y}_i \}, i = 1 \ldots P.
\tag{5}
$$

By using the algorithm KFCM-K for $\mathbf{R_X}$, a CDS of *C* data fuzzy clusters signed $\Gamma^k$, $k = 1 \ldots C$, is obtained.

**Definition 1.** *(Convergent degree). Convergent degree $d_{ak}$ of $\Gamma^k$ is defined as follows:*

$$
d_{ak} \; (k = 1 \ldots C) = \begin{cases}
1 - (\widetilde{p} - p_k)/\widetilde{p} \text{ if } \widetilde{p} > p_k \\
1 \text{ if } \widetilde{p} \leq p_k,
\end{cases}
\tag{6}
$$

*where $\widetilde{p} = P/C$, and $p_k$ is the number of data points to be distributed hard in $\Gamma^k, k = 1 \ldots C$.*

**Definition 2.** *(Distribution radius). Distribution radius of data cluster $\Gamma^k$, signed $R_{A(k)}$ is defined:*

$$
R_{A(k)} = \left( \frac{1}{p_k} \sum_{q=1}^{P} \overline{\mu}_{kq}(\overset{-}{\mathbf{x}}_q) \left\| \overset{-}{\mathbf{x}}_q - \overset{-0}{\mathbf{x}}_k \right\|^2 \right)^{0.5}, \; k = 1 \ldots C,
\tag{7}
$$

*where $\mu_{kq}(\overline{\mathbf{x}}_q)$ is the membership degree of the q-th data point belonging to the k-th cluster to be estimated by (4); $\left\|\overline{\mathbf{x}}_q - \overline{\mathbf{x}}_k^0\right\|$ is the Euclidean distance between data point $\overline{\mathbf{x}}_q$ and cluster centroid $\overline{\mathbf{x}}_k^0$. The normalized distribution radius $R_k$ is defined as follows:*

$$R_k = R_{A^{(k)}} / \max_h \left( R_{A^{(h)}} \right), \ k = 1 \ldots C, h = 1 \ldots C. \tag{8}$$

**Remark 1.** *The distribution radius $R_k$ in Definition 2 is normalized to remove the dimension. Based on this approach, the filter FIN proposed in Section 3 can match with different numerical databases well.*

**Definition 3.** *(Data dispersion). The dispersion of data points in the k-th cluster signed $d_k$ is then defined:*

$$d_k = R_k \tanh \left( \eta R_k d_{ak}^{-1} \right), \ k = 1 \ldots C, \tag{9}$$

*where $1 < \eta \leq \overline{\eta} = 0.5\pi C / \sum_{k=1}^{C} R_k d_{ak}^{-1}$ is an experience formula.*

2.2.2. The AfODST

IDS : $\left( \mathbf{R_X} \in \Re^{P \times (n+1)}, \mathbf{y} \in \Re^{P \times 1} \right)$ can be redepicted

$$\langle \text{input} - \text{output} \rangle = \langle \mathbf{R_X} - \mathbf{y} \rangle \tag{10}$$

From a given IDS, ODST-trainset and ODST-testset are set up. They are the two distinct datasets with the same size and input–output structures as shown in (10). Then, an ANFIS named ANFIS_train (with $C_1$ input data clusters in the input cluster data space iCDS$_1$) is built from ODST-trainset; also, an ANFIS_test (with $C_2$ input data clusters in the input cluster data space iCDS$_2$) is set up based on ODST-testset. In these works, the algorithm ANFIS-JS is adopted for establishing CDSs. From iCDS$_1$, $d_k$, $k = 1 \ldots C_1$, (9) is estimated. One then (i) seeks the m-th cluster satisfying $d_m = \max\limits_{i=1 \ldots C_1} d_i$, (ii) removes this hard cluster along with all the data points hard distributed in it, and (iii) updates the ANFIS_train net based on the remaindered clusters in iCDS$_1$. The result obtained is the updated ANFIS_train net denoted uANFIS_train, filtered ODST-trainset denoted fODST-trainset, and $C_1 := C_1 - 1$. Similarly, one gets uANFIS_test from ANFIS_test, fODST-testset from ODST-testset, and $C_2 := C_2 - 1$. The input of fODST-testset is eventually used for uANFIS_train to calculate the error, Equation (11), where $\hat{y}_j$ is the j-th output of uANFIS_train.

$$\text{E} = \left( \sum_{j=1}^{P} \left( \hat{y}_j - y_j \right)^2 / P \right)^{0.5}. \tag{11}$$

This loop process is carried out until $\text{E}(h) \approx \text{E}(h-1)$. As a result, the ODST is the value of $d_m = \max\limits_{i=1 \ldots C_1} d_i$ at the $(h-1)$-th loop. This content can be depicted via the two procedures and Algorithm 1 below.

**Procedure 1.** *(For probing and removing IN). At the h-th loop, calculate $d_k \equiv d_k(h)$ using Equation (9) for iCDS$_1$ ($C_1$ clusters) and iCDS$_2$ ($C_2$ clusters). For iCDS$_1$: look for the cluster satisfying $d_{m1}(h) = \max\limits_{k=1 \ldots C_1} d_k(h)$; cancel this cluster and its data points, and restructure ANFIS_train; fODST-trainset, uANFIS_train, and $C_1 := C_1 - 1$ are the obtained results. For iCDS$_2$: similarly, look for the cluster satisfying $d_{m2}(h) = \max\limits_{k=1 \ldots C_2} d_k(h)$, cancel this cluster along with its data points, and restructure ANFIS_test; fODST-testset, uANFIS_test, and $C_2 := C_2 - 1$ are the ones coming from this phase.*

**Procedure 2.** *(for quantifying the ODST). The input of fODST-testset is used for the uANFIS_train to calculate $\text{E}(h)$ (11). The loop is to be continued if $\text{E}(h) < \text{E}(h-1)$. Otherwise, if $\text{E}(h) \approx$*

$E(h-1)$: *stop this process and fix* ODST $:= d_{m1}(h-1)$. $C_1$ *corresponding to this loop is called* $C_1^{finish}$.

---

**Algorithm 1** The AfODST (Nguyen et al. [19])

---

*Input:* IDS : $\left( \mathbf{R_X} \in \Re^{P \times (n+1)}, \mathbf{y} \in \Re^{P \times 1} \right)$ (10)

*Output:* The ODST of the IDS (together with [E], [$h$], $C_1^{begin}$ and $C_1^{finish}$)

1. Build ODST-trainset and ODST-testset.
2. From ODST-trainset, ODST-testset and [E]: set up ANFIS_train (iCDS1, $C_1$ clusters) and ANFIS_test (iCDS2, $C_2$ clusters).
3. Determine the ODST: ODST $:= 0$; $h := 0$

   **While** ODST = 0 and $h \leq [h]$
       $h := h + 1$; at the $h$-th loop:
       (a) Perform **Procedure 1**
       (b) Perform **Procedure 2**
   **End While**
       (c) Save the ODST, [E], [$h$], $C_1^{begin}$ and $C_1^{finish}$; Stop.

---

## 3. Proposed Method of DDM-Based Fault Diagnosis

### 3.1. Proposed Filter FIN for Cancelling IN

Based on Algorithm 1 to quantify the ODST, the ODST-based filter named ODSTbF has been presented in [19]. However, its application scope is only the databases describing the dynamic response of smart dampers, such as MR or ER dampers. In this subsection, we propose a new filter called FIN for filtering IN in any mechanical vibration database in the form (10) for the fault diagnosis of mechanical systems. The FIN's approach relies on two vital observations, as follows.

1. The first observation. For DDM-based mechanical-system fault diagnosis, in general, a labeled source domain $(\mathbf{X}^s, \mathbf{y^s})$ is built to describe the dynamic response of the system in the training phase while a target domain $\mathbf{X}^t$ is set up in the system's operating process that provides dynamic response information of the system at the survey time, where $\mathbf{X}^s$ and $\mathbf{X}^t$ are matrixes of data samples and $\mathbf{y^s}$ is the label vector of $\mathbf{X}^s$. The fact that there are labeled samples $_l\mathbf{X}^t$ (their label is $_l\mathbf{y}^t$) in $\mathbf{X}^t$ corresponding to the managed object's healthy status, then the remainder $_u\mathbf{X}^t$ ($_u\mathbf{X}^t \cup {}_l\mathbf{X}^t = \mathbf{X}^t$) is the unlabeled samples. The fault identification here is to classify/label $_u\mathbf{X}^t$ based on the labeled samples.

2. The second observation. Although a domain drift between the source and target domains always exists, the difference between the data correlation in the database $(\mathbf{X}^s \cup \mathbf{X}^t)$ without noise compared with that with INs can be recognized via their data distribution status.

Let us consider a data matrix $\mathbf{X}$ with noise satisfying the three following aspects. (i) It derives from source/target domain of the surveyed mechanical system. (ii) It is in the form of $_l\mathbf{X}^t$ such as $_u\mathbf{X}^t$ (including $_l\mathbf{X}^t$) or the individual parts in $\mathbf{X}^s$ deriving from a certain single fault status. (iii) Its size must be equal to the size of $_l\mathbf{X}^t$. The proposed FIN filter for removing noise in $\mathbf{X}$ is then described in Algorithm 2 below.

---

**Algorithm 2 Filter FIN**

---

*Input:* $(\mathbf{X}^s, \mathbf{y^s})$ and $({}_l\mathbf{X}^t, {}_l\mathbf{y}^t)$ deriving from the managed mechanical system.
A measured data matrix $\mathbf{X}$ with noise as described above
*Output:* The filtered $\mathbf{X}$

1. Set up an input-output dataset in the form of (10) named IDS_1 (12) where $\tilde{\mathbf{y}}^s$ and ${}_l\tilde{\mathbf{y}}^t$ respectively are the data vectors normalized from $\mathbf{y}^s$ and ${}_l\mathbf{y}^t$.

2. Determine the ODST of the IDS_1 based on Algorithm 1 such that (i) the ODST-trainset is selected to be equal to the IDS_1, and (ii) although the ODST-testset is different from the ODST-trainset, their sizes are similar (see Section 2). (Obtaining: the ODST of the IDS_1 along with Z, $C_1^{begin}$ and $C_1^{finish}$)

3. Establish an input-output dataset named IDS_2 (12) that is in the form of the IDS_1 but contains $\mathbf{X}$.

4. Filter INs in $\mathbf{X}$

    (a) From the IDS_2, build a cluster data space of the clusters $\Gamma^k, k = 1 \ldots C_1^{finish}$ by initializing $C = C_1^{begin}$ and using the algorithm KFCM-K (Marcelo et al. [20])

    (b) Calculate $d_k$ from Equation (9) for $\Gamma^k, k = 1 \ldots C_1^{finish}$, cancel all the clusters (and their data points) satisfying $d_k >$ ODST, and update the data matrix $\mathbf{X}$.

---

$$\begin{cases} \text{IDS\_1} \equiv \langle \text{input} - \text{output} \rangle_{(1)} = \left\langle \left( \mathbf{R_{X1}} = (\mathbf{X}^s \cup \tilde{\mathbf{y}}^s) \cup ({}_l\mathbf{X}^t \cup {}_l\tilde{\mathbf{y}}^t) \right) - \left( \mathbf{y}_{(1)} = \mathbf{y}^s \cup {}_l\mathbf{y}^t \right) \right\rangle \\ \text{IDS\_2} \equiv \langle \text{input} - \text{output} \rangle_{(2)} = \left\langle \left( \mathbf{R_{X2}} = (\mathbf{X}^s \cup \tilde{\mathbf{y}}^s) \cup (\mathbf{X} \cup {}_l\tilde{\mathbf{y}}^t) \right) - \left( \mathbf{y}_{(1)} = \mathbf{y}^s \cup {}_l\mathbf{y}^t \right) \right\rangle \end{cases} \tag{12}$$

**Remark 2.** *(i) It can be inferred from the first observation that the way of organizing data in IDS_1 and IDS_2 (12) enriches information related to the labeled data samples covering both domains. This approach allows not only to weaken negative influence of the domain drift between the source and target domains, but also to increase the difference of data correlation between $(\mathbf{X}^s \cup \mathbf{X}^t)$ with IN and $(\mathbf{X}^s \cup \mathbf{X}^t)$ without IN. It is meaningful to improve the filtering effectiveness, and also allow for exploiting the ODST (obtained in Step 2) for $\mathbf{X}$ (Step 4) deriving from any domain, not only the source domain but also the target domain. (ii) In (12), operator $\cup$ in $(\mathbf{X}^s \cup \tilde{\mathbf{y}}^s)$, $({}_l\mathbf{X}^t \cup {}_l\tilde{\mathbf{y}}^t)$, and $(\mathbf{X} \cup {}_l\tilde{y}^t)$ increases the data dimension (a total of the two data dimensions) without increasing the number of samples; while $\cup$ in $(.) \cup (.)$ and $\mathbf{y}^s \cup {}_l\mathbf{y}^t$ increases the number of data samples (the total of the sample numbers) without increasing the data dimension.*

### 3.2. Building Databases for the BFDM

### 3.2.1. A Multi-Feature

From six the selected single features deriving from Wu et al. [21], we set up a multi-feature (*MF*) (13) from the acceleration-sensors-based signal $X(t_i)$:

$$MF(k) = \begin{bmatrix} X_{rmsv} = \left( (1/N)\sum_{i=1}^{N} X^2(t_i) \right)^{0.5} \\ X_{mav} = \max(X(t_i)) \\ X_{smrv} = \left( (1/N)\sum_{i=1}^{N} \sqrt{X(t_i)} \right)^2 \\ X_{kc} = \frac{1}{X_{rms}^4}\sum_{i=1}^{N}\left( X(t_i) - \frac{1}{N}\sum_{k=1}^{N} X(t_k) \right)^4 \\ X_{cf} = X_{mav}/X_{rmsv} \\ X_{rmsf} = \left( \sum_{i=2}^{N} \dot{X}^2(t_i)/4\pi^2\sum_{i=1}^{N} X^2(t_i) \right)^{0.5} \end{bmatrix}, \tag{13}$$

where $t_i$ is the $i$-th sampled time and $N$ is the number of sampling points.

In (13), amplitude and energy are reflected by RMSV, MAV, and SMRV, the data's distribution situation is expressed via KC and CF, while the RMSF provides the signal's varying speed. Another approach, paying attention to analyzing the signal in the time–frequency domain, was presented by Truong et al. [22]. In this study, we propose a way of

exploiting the RMSV, MAV, SMRV, KC, CF, and RMSF in the time–frequency domain based on the method presented by Truong [22] and others. Its advantage is to enrich the obtained information by considering the single features in individual frequency ranges. The next subsection details these aspects.

### 3.2.2. Building Databases

From measurements, we have $H$ vibration datasets (14) in the time domain related to $H$, the considered fault statuses:

$$[\mathbf{D}_1, \ \mathbf{D}_2, \ldots, \mathbf{D}_H]^T \tag{14}$$

For $\mathbf{D}_i (1 \leq i \leq H)$, based on SSA we obtain the $p$ time series corresponding to different frequency ranges as in (15):

$$[\mathbf{D}_{i1}, \ \mathbf{D}_{i2}, \ldots, \mathbf{D}_{ip}], \ i = 1 \ldots H \tag{15}$$

Note that $p$ in (15) is the parameter selected by the designer. Because the mechanical vibration signal is prone to the low-frequency range [11], by placing the eigenvalues in decreasing order, among the $p$ subsets, we cancel $(p\text{-}q)$, the last subsets related to the high-frequency signal ranges. The remainder time series are employed to build the database as:

$$[\mathbf{D}_{i1}, \ \mathbf{D}_{i2}, \ldots, \mathbf{D}_{iq}], \ i = 1 \ldots H \tag{16}$$

The mechanical vibration signal analyzed into frequency-based individual series as in (16) is then used to describe the source and target domains. Namely, the input space of the source domain is illustrated by the matrix (17), where $L$ is the length of vector $\mathbf{D}_{ik}, k = 1 \ldots q$, while "$R$" means the raw data to be exploited.

$$\mathbf{X}^s{}_R = \begin{pmatrix} \mathbf{D}_{11} & \cdots & \cdots & \mathbf{D}_{1q} \\ \vdots & \ddots & \ddots & \vdots \\ \vdots & \ddots & \ddots & \vdots \\ \mathbf{D}_{P_01} & \cdots & \cdots & \mathbf{D}_{P_0q} \end{pmatrix} \in \Re^{P_0 \times q}, P_0 = LH. \tag{17}$$

Note that the health status of the managed mechanical system is not changed significantly in a survey cycle. In this cycle, therefore, we can set up the input space of the labeled and unlabeled raw-data parts of the target with the same length $L$, as follows:

$$_l\mathbf{X}^t_R = \begin{pmatrix} {}_l\mathbf{D}_{11} & \cdots & \cdots & {}_l\mathbf{D}_{1q} \\ \vdots & \ddots & \ddots & \vdots \\ \vdots & \ddots & \ddots & \vdots \\ {}_l\mathbf{D}_{L1} & \cdots & \cdots & {}_l\mathbf{D}_{Lq} \end{pmatrix} \in \Re^{L \times q}; \ _u\mathbf{X}^t_R = \begin{pmatrix} {}_u\mathbf{D}_{11} & \cdots & \cdots & {}_u\mathbf{D}_{1q} \\ \vdots & \ddots & \ddots & \vdots \\ \vdots & \ddots & \ddots & \vdots \\ {}_u\mathbf{D}_{L1} & \cdots & \cdots & {}_u\mathbf{D}_{Lq} \end{pmatrix} \in \Re^{L \times q}. \tag{18}$$

For the output, the $j$-th fault type is encoded via a certain real number $y_j$. Thus, vectors $\mathbf{y}^s$ and $_l\mathbf{y}^t$ are yielded, which depict the output of the source and the labeled part in the target domain, respectively.

Subsequently, filtering IN in $\mathbf{X}^s{}_R$ or $_l\mathbf{X}^t_R$ or $_u\mathbf{X}^t_R$ is performed based on the FIN (see Section 3.1). These filtered databases are redenoted $\mathbf{X}^s$, $_l\mathbf{X}^t$, and $_u\mathbf{X}^t$, respectively. Building datasets for training, testing, and checking rely on them. Specifically, by sliding a window (with a width of $N$ data points) along the columns of the $\mathbf{X}^s$, or $_l\mathbf{X}^t$ or $_u\mathbf{X}^t$, such that data for calculating one value of $MF(.)$ must derive from a health's single status of the managed object, we obtain feature matrixes in the form of $\mathbf{M}(.)$ in (19):

$$\mathbf{M}(.) = \begin{pmatrix} MF(1,1) & MF(1,2) & \cdots & \cdots & MF(1,kq) \\ MF(2,1) & MF(2,2) & \cdots & \cdots & MF(2,kq) \\ \vdots & \vdots & \ddots & \ddots & \vdots \\ MF((\widetilde{P}-1),1) & MF((\widetilde{P}-1),2) & \ddots & \ddots & MF((\widetilde{P}-1),kq) \\ MF(\widetilde{P},1) & MF(\widetilde{P},2) & \cdots & \cdots & MF(\widetilde{P},kq) \end{pmatrix} \in \Re^{\widetilde{P}\times(kq)}, \tag{19}$$

where $k = 6$ is the number of the single features in (13). Let $\widetilde{P}_0$ be the number of the established data samples corresponding to only one health's single status of the managed object, then $\widetilde{P} = \widetilde{P}_0$ for $\mathbf{M}(.)$ deriving from either the $_l\mathbf{X}^t$ or $_u\mathbf{X}^t$, while $\widetilde{P} = \widetilde{P}_0 H$ for $\mathbf{M}(.)$ deriving from the $\mathbf{X}^s$.

To identify the dynamic response of the mechanical system where the managed object is installed, we build two datasets, one named TrainS for training and the other named TestS for testing. The form of the TrainS and TestS are shown in (20), in which IDS and ODS, respectively, denote the input and output data space:

$$\text{database} \equiv [\text{IDS} - \text{ODS}] \equiv \left[\mathbf{M}(\mathbf{X}^s \cup {}_l\mathbf{X}^t) - \overline{\mathbf{y}}\right], \tag{20}$$

where $\mathbf{M}(\mathbf{X}^s \cup {}_l\mathbf{X}^t) \in \Re^{(H+1)\widetilde{P}_0\times n}$ comes from (19) and $\overline{\mathbf{y}} \in \Re^{P\times 1}$ comes from (21), $P = \widetilde{P}_0(H+1)$, $n = kq$:

$$\overline{\mathbf{y}} \equiv \mathbf{y}_{(1)} = \left[\mathbf{y}^s \cup {}_l\mathbf{y}^t\right]^T = $$
$$\left[{}^1\mathbf{y} \in \Re^{1\times\widetilde{P}_0},\ldots,{}^H\mathbf{y} \in \Re^{1\times\widetilde{P}_0}, {}_l\mathbf{y}^t \in \Re^{1\times\widetilde{P}_0}\right]^T \in \Re^{P\times 1}, \tag{21}$$

In (21), ${}^t\mathbf{y} \in \Re^{1\times P_0}$ corresponds to the $t$-th fault type, $t = 1\ldots H$.

In fact, the two parameters, $L$ and $q$ in (17) and (18), need to be optimized. Note that their optimal values are understood as their values such that the fault diagnosis effectiveness of the ANFIS-BFDM is the best. Because the efficiency of the ANFIS-BFDM is reflected by the mean accuracy (*MeA*) (23), so here we adopt the objective function (22) to optimize $L$ and $q$:

$$J(L,q) \equiv MeA(L,q) \to \max \tag{22}$$

$$MeA = \frac{100 \times \sum_{h=1}^{H} true\_samples_h}{\sum_{h=1}^{H} total\_samples_h}(\%) \tag{23}$$

where $h$ denotes the $h$-th damage type, $h = 1 \ldots H$, $true\_samples_h$ is the number of checking samples expressing correctly the real status of the bearing(s), while $total\_samples_h$ is the total of the checked samples.

### 3.3. The ANFIS-BFDM

The proposed ANFIS-BFDM consists of two phases described in Algorithm 3 below. The first phase is an offline phase to identify the dynamic response of the mechanical system where the bearing(s) is installed. The second phase is an online phase to estimate online the bearing(s) health status at each checking time. They are described as follows:

---

**Algorithm 3** ANFIS-BFDM

---

*Input:* Measured data $\mathbf{X}^s$ and $\mathbf{X}^t$ with noise in the source and target domains.
*Output:* The $\mathbf{X}^t$ to be labeled to reflect the health status of the managed bearing(s).
***In the offline phase:***
Initialize $L$, $q$, and $H$ in Equations (16) and (17).

1. Measure to provide time raw-data series of source and target domains.
2. Build/rebuild the raw input $\mathbf{X}^s_R(L, q)$ (17), $_l\mathbf{X}^t_R(L, q)$ (18) and the output vectors $\mathbf{y}^s$ and $_l\mathbf{y}^t$.
3. Filter IN in $\mathbf{X}^s_R(L, q)$ and $_l\mathbf{X}^t_R(L, q)$ to obtain $\mathbf{X}^s(L, q)$ and $_l\mathbf{X}^t(L, q)$ using the filter FIN (Section 3.1), then save the key parameters of the FIN to be the ODST, $C_1^{begin}$ and $C_1^{finish}$.
4. Identify the dynamic response of the mechanical system installed the managed bearing(s) via the $\mathbf{X}^s_R(L, q)$ and $_l\mathbf{X}^t_R(L, q)$:

   (a) Set up two different datasets TrainS and TestS in the form of (20);
   (b) Train an ANFIS using TrainS and the algorithm ANFIS-JS (Nguyen et al. [16]);
   (c) Optimize $L$ and $q$ upon the DE (differential evolution) (Gong and Cai [23]) and a loop process from Step 2 to Step 4 until the function (22) is minimized. In this process, TestS is used as the input–output dataset to test the difference between the output of the ANFIS and the data output.

***In the online phase:***

5. Measure to build a raw database $_u\mathbf{X}^t_R$ (18).
6. Filter IN in the $_u\mathbf{X}^t_R$ to obtain the $_u\mathbf{X}^t$ upon the FIN with its parameters coming from Step 3.
7. Set up a dataset named CheckS from the $_u\mathbf{X}^t$ that is in the form of $\mathbf{M}(.)$ (19), $\mathbf{M}(.) = \mathbf{M}(_u\mathbf{X}^t) \in \Re^{\widetilde{P}_0 \times n}$.
8. Classify the CheckS to recognize the bearing(s) health status as follows:

   (a) Employ the CheckS as the input of the ANFIS to obtain its output $\hat{\mathbf{y}}$.
   (b) The label of the CheckS is $d$ such that Equation 24 below is satisfied.

---

$$\left\| \hat{\mathbf{y}} - {}^d\mathbf{y}^s \right\| = \min_{i=1\ldots H} \left\| \hat{\mathbf{y}} - {}^i\mathbf{y}^s \right\| \tag{24}$$

## 4. Evaluating the ANFIS-BFDM

### 4.1. Establishing Database

We use two data sources of bearing acceleration to evaluate the proposed ANFIS-BFDM in the three cases, as in Table 2. Data Source 1 derives from Case Western Reserve University "12k Drive End Bearing Fault Data", "DE" (drive end accelerometer data), the fault diameter D1 = 0.014, and the motor load Lk (L0 = 0, L1 = 1HP) (https://csegroups.case.edu/bearingdatacenter/pages/download-data-file) (accessed on 7 July 2021). Data Source 2 is measured upon our experimental apparatus detailed in Figure 1, in which the damage degree (n) and its location are provided at the end of Table 2. In these tables, BaDnL*m*, InDnL*m*, and OuDnL*m*, respectively, define load degree to be m, damage level to be n, and damage location to be at the Ball, or Inner, or Outer of the bearing; NM shows the bearing to be undamaged.

**Table 2.** Three surveyed data cases and crack sizes in Case 3.

| Three Surveyed Data Cases | | |
|---|---|---|
| **Case 1 (Data Source 1)** | **Case 2 (Data Source 1)** | **Case 3 (Data Source 2)** |
| NML0 | NML1 | NML0 |
| BaD1L0 | BaD1L1 | BaD1L0 |
| InD1L0 | InD1L1 | InD1L0 |
| OuD1L0 | OuD1L1 | OuD1L0 |
| **Crack Size in Case 3** | | |
| **Fault Degrees and Their Location** | **Width (mm)** | **Depth (mm)** |
| BaD1 | 0.15 | 0.2 |
| InD1 | 0.2 | 0.3 |
| OuD1 | 0.2 | 0.3 |

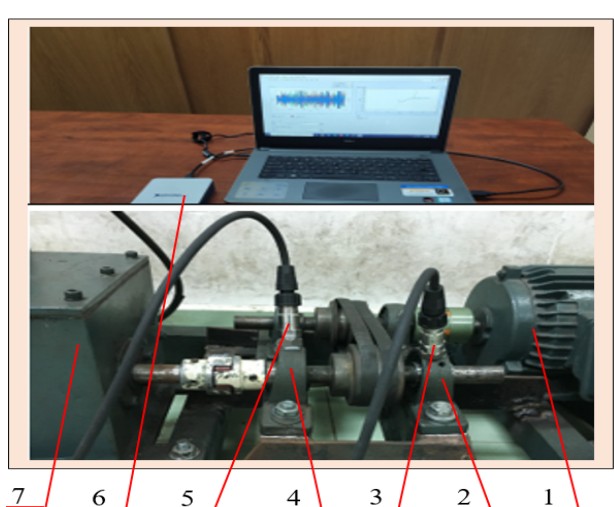

**Figure 1.** Experimental apparatus for measuring vibration signal: motor (**1**), acceleration sensors (**3**) and (**5**), the surveyed bearings (**2**) and (**4**), the module for processing and transforming series vibration signal (Model: NI-9234) (**6**), the gearbox and the brake used for changing load (**7**).

*4.2. Estimation Method and Results Obtained*

Together with the *MeA* (23) we also employ the root means square error (*RMSE*) (25) to verify the effectiveness:

$$RMSE = \sqrt{\sum_{i=1}^{P} (y_i - \hat{y}_i)^2 / P}, \tag{25}$$

where $y_i$ and $\hat{y}_i$, respectively, are the encoding and predicting outputs.

We carry out surveys based on the three databases, given in Table 2. In each case, we select the number of samples to be 2000. Additionally, the optimal value of $q$ for these three databases is three, as presented below. As a result, due to $\tilde{P}_0 = 2000$, $q = 3$, $H = 4$, and $k = 6$, we obtain $P = 10{,}000$, $n = 18$ (see Equations (20) and (21)) along with the IDS and ODS of the TrainS and TestS to be IDS $\in \Re^{10{,}000 \times 18}$ and ODS $\in \Re^{10{,}000 \times 1}$ (for setting up the ANFIS in Step 4); while the IDS of the CheckS is IDS $\in \Re^{2000 \times 18}$ (for recognizing the bearing's health status in Step 8 of the ANFIS-BFDM).

To increase the severity of impulse noise in the surveys, we include with the measured data an impulse source called $r(t)$ whose pulse intensity does not exceed 2.4 times the amplitude of the data such that 5% of the measured data points are impacted by this source. For this work, each aforementioned random noise signal is added to a certain data point belonging to 5% of the database randomly but not repeatedly. Figure 2 illustrates the participation of $r(t)$ in the InD1L1 belonging to Case 2, shown in Table 2.

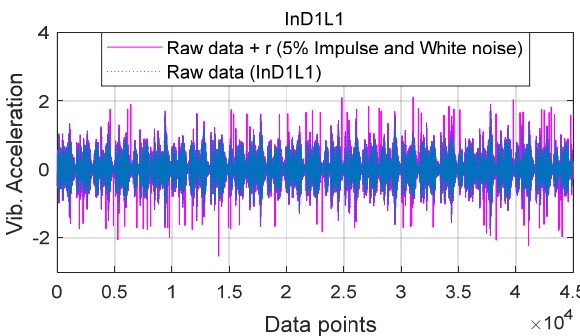

**Figure 2.** An example of the noise part *r(t)* participating in the InD1L1 belonging to the dataset "Case 2" (see Table 2).

As mentioned above, the filter FIN is improved from [19] to manage mechanical vibration databases. Because the FIN takes a vital part in the proposed ANFIS-BFDM, we consider the positive role of the FIN and the effectiveness of the ANFIS-BFDM in their mutual relationship via the *RMSE* (25) and *MeA* (23). Accordingly, the answer to the question concerning the accuracy of the built testing and learning datasets can be inferred through the *RMSE* and *MeA* of the results obtained from ANFIS-BFDM in two cases, using the FIN and not using the FIN. Regarding the comparison results, we analyze and compare the fault diagnosis results of the ANFIS-BFDM with/without the FIN and the corresponding results deriving from some other methods of bearing diagnosis. They are (i) the IFDUFL (intelligent fault diagnosis using unsupervised feature learning, [10]); (ii) the BDIM (online bearing damage identifying method based on ANFIS, SSA, and sparse filtering, [11]); (iii) the CHMM (continuous hidden Markov model for diagnosing bearing fault, [17]), and (iv) the AfOBSM (algorithm for building a system of online bearing status monitoring, [22]). The importance of optimizing the ANFIS-BFDM's parameters consisting of *q* and *L* in Equation (18) and the method for their optimal quantification are also discussed and addressed through these surveys.

The results obtained from the surveys are shown in Figures 3–5 and Tables 1–5.

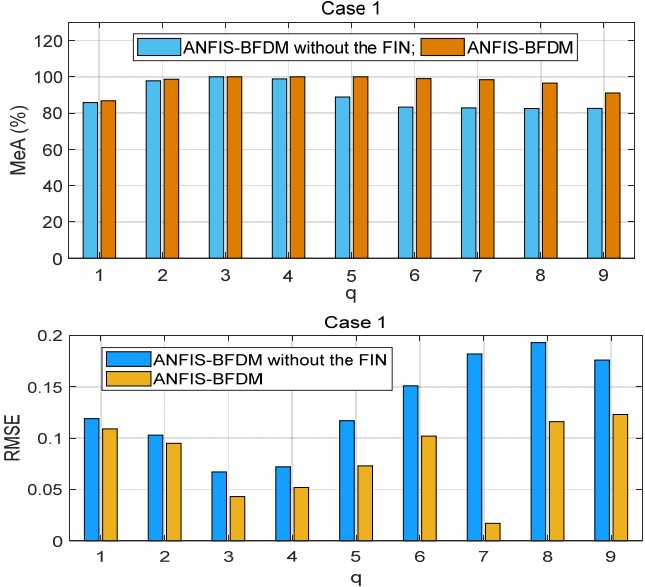

**Figure 3.** Deriving from the dataset "Case 1" (see Table 2)—the positive role of the filter FIN reflected via the *MeA* and *RMSE* of the ANFIS-BFDM in two cases, with/without *r(t)* when *q* (in Equation (16)) verifying from 1 to 9.

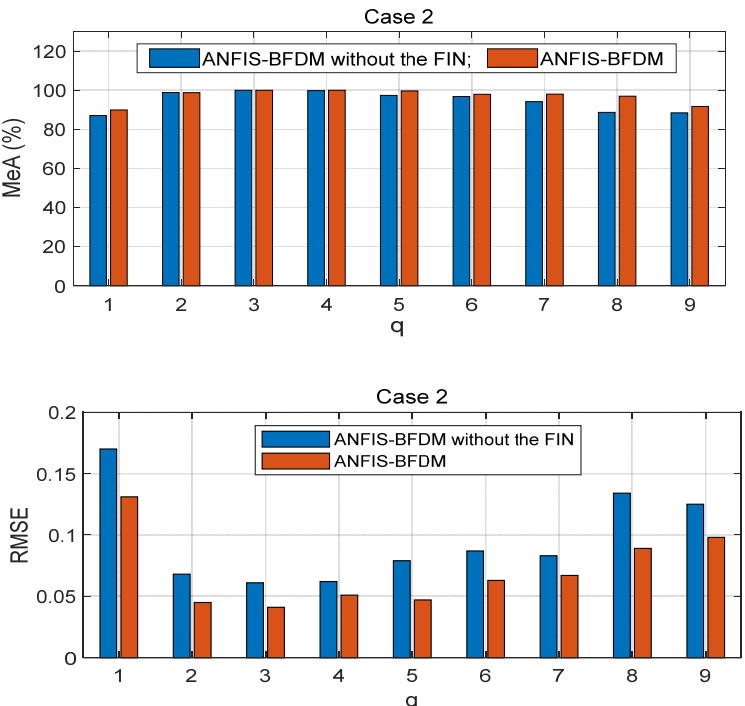

**Figure 4.** "Case 2"—the positive role of the FIN expressed by the *MeA* and RMSE of the ANFIS-BFDM via the two cases, with/without *r(t)* when *q* = 1 . . . 9.

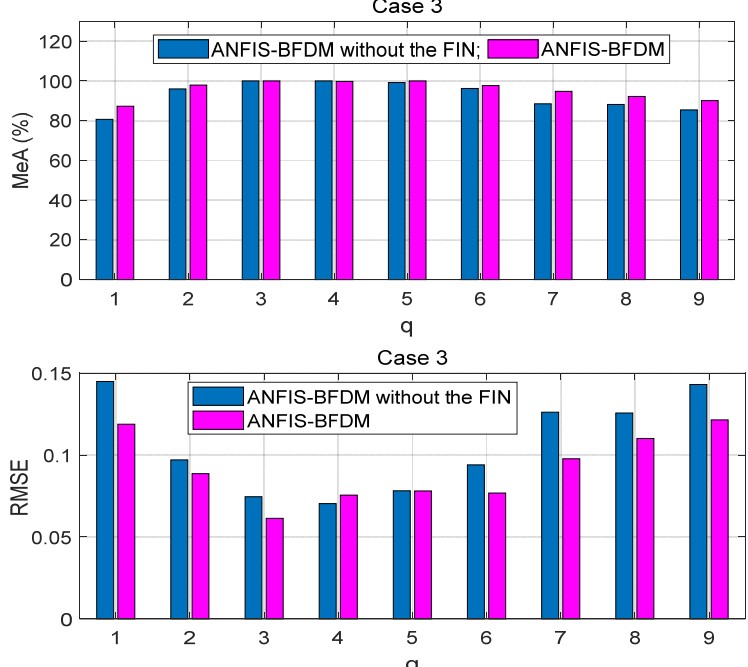

**Figure 5.** Deriving from the dataset "Case 3" (see Table 2)—the positive role of the filter FIN reflected via the *MeA* and *RMSE* of the ANFIS-BFDM in two cases, with/without *r(t)* when *q* = 1 . . . 9.

**Table 3.** The *MeA* (%) of the ANFIS-BFDM in the three data cases (see Table 2) depends on $q$ (16) (WF: with the FIN; WOF: without the FIN; the bold: the results when $q = q_{opt} = 3$).

| $q$ | Case 1 | | Case 2 | | Case 3 | |
|---|---|---|---|---|---|---|
| | **WOF** | **WF** | **WOF** | **WF** | **WOF** | **WF** |
| 1 | 85.83 | 86.76 | 87.08 | 89.96 | 80.69 | 87.24 |
| 2 | 97.78 | 98.68 | 98.89 | 98.76 | 95.97 | 97.90 |
| **3** | **100** | **100** | **100** | **100** | **100** | **100** |
| 4 | 98.89 | 100 | 99.86 | 100 | 100 | 99.76 |
| 5 | 88.89 | 100 | 97.36 | 99.67 | 99.17 | 100 |
| 6 | 83.33 | 99.01 | 96.81 | 97.98 | 96.25 | 97.65 |
| 7 | 82.92 | 98.45 | 94.17 | 98.09 | 88.47 | 94.76 |
| 8 | 82.50 | 96.61 | 88.75 | 97.00 | 88.19 | 92.19 |
| 9 | 82.64 | 91.08 | 88.47 | 91.75 | 85.42 | 90.12 |

**Table 4.** The *RMSE* of the ANFIS-BFDM in the three data cases (see Table 2) depends on $q$ (16) (WF: with the FIN; WOF: without the FIN; the bold: the results when $q = q_{opt} = 3$).

| $q$ | Case 1 | | Case 2 | | Case 3 | |
|---|---|---|---|---|---|---|
| | **WOF** | **WF** | **WOF** | **WF** | **WOF** | **WF** |
| 1 | 0.119 | 0.109 | 0.170 | 0.131 | 0.145 | 0.119 |
| 2 | 0.103 | 0.095 | 0.068 | 0.045 | 0.097 | 0.089 |
| **3** | **0.067** | **0.043** | **0.061** | **0.041** | **0.075** | **0.061** |
| 4 | 0.072 | 0.052 | 0.062 | 0.051 | 0.070 | 0.076 |
| 5 | 0.117 | 0.073 | 0.079 | 0.047 | 0.078 | 0.078 |
| 6 | 0.151 | 0.102 | 0.087 | 0.063 | 0.094 | 0.077 |
| 7 | 0.182 | 0.017 | 0.083 | 0.067 | 0.126 | 0.098 |
| 8 | 0.193 | 0.116 | 0.134 | 0.089 | 0.126 | 0.110 |
| 9 | 0.176 | 0.123 | 0.125 | 0.098 | 0.143 | 0.122 |

**Table 5.** $L_{opt}$ corresponding to the single faults.

| Source 1 | $L_{opt}$ | Source 2 | $L_{opt}$ |
|---|---|---|---|
| NML0/L1 | 31,600 | NML0 | 49,900 |
| BaD1L0/L1 | 53,800 | BaD1L0 | 68,500 |
| InD1L0/L1 | 35,600 | InD1L0 | 51,340 |
| OuD1L0/L1 | 35,500 | OuD1L0 | 51,500 |

*4.3. Discussion*

Some vital points drawn from Figures 3–5 and Tables 3–6 are as follows.

**Table 6.** Comparing results related to the three datasets in two cases: with/without noise $r(t)$.

| Methods | MeA (%) | | | | | |
|---|---|---|---|---|---|---|
| | **Case 1** | **Case 2** | **Case 3** | **Case 1 with $r$** | **Case 2 with $r$** | **Case 3 with $r$** |
| IFDUFL | 100 | 99.54 | 95.82 | 94.34 | 92.11 | 90.56 |
| BDIM | 96.98 | 100 | 100 | 93.18 | 92.87 | 91.01 |
| CHMM | 92.19 | 94.25 | 94.76 | 87.56 | 89.93 | 84.65 |
| AfOBSM | 96.82 | 98.36 | 94.55 | 90.13 | 93.11 | 89.73 |
| ANFIS-BFDM | **100** | **100** | **100** | **96.87** | **96.12** | **94.99** |

The positive role of the filter FIN for the proposed algorithm ANFIS-BFDM is reflected clearly by the results in Tables 3 and 4 and Figures 3 and 5. In all the surveyed databases, the MeAs (23) of the ANFIS-BFDM with the FIN are almost higher than that without the

FIN; also, the RMSEs (25) of the proposed method with FIN are always lower than that without the FIN.

The effectiveness of the ANFIS-BFDM depends significantly on the parameter $q$ in Equation (16). The appropriate selection of this parameter (denoted $q_{opt}$), therefore, is very critical. It relates to the reconstruction of SSA and is one of the two necessary conditions for maximizing the objective function (22). Note that $q_{opt}$ depends on many factors such as the mechanical characteristics of the health-managed object, its operation condition, and especially the noise status coming from uncertainties that could not be estimated fully to filter completely. Therefore, a sufficiently wide range of $q$ must be employed when looking for $q_{opt}$ to avoid unsatisfactory conclusions. For example, in Case 3 with $q = 4$, shown in Tables 3 and 4, the ANFIS-BFDM with the FIN provided even worse results than the ANFIS-BFDM without the FIN.

The stability of $q_{opt}$ when the operating condition of the mechanical systems is to be changed can be recognized from the survey results in Tables 3 and 4 or Figures 3–5, namely, based on the MeA in Table 3: for the dataset "Case 1", $q_{opt}$ is 3 or 4 or 5; for "Case 2", $q_{opt}$ is 3 or 4; for "Case 3", $q_{opt}$ is 3 or 5. Based on the RMSE in Table 4, for all cases, $q_{opt}$ is 3. Hence, we select $q = 3$ for all these datasets.

Note that finding the optimal value ($L_{opt}$) of $L$ in (17) is necessary to ensure the advantages of the proposed method. Meaningful information about the health status of the mechanical system is lost if $L$ is less than $L_{opt}$. Conversely, the calculating cost rises if $L$ is much higher than $L_{opt}$. The survey results obtained from the single damages are shown in Table 5. As a result, the optimal values of $L$ for datasets combined from single faults are the maximum value of the constituent ones. For example,

$$L_{opt}(NML0/L1, BaD1L0/L1) = 34800. \tag{26}$$

Finally, the compared results illustrated in Table 6 reflect that, although impacted by noise, the ability of the proposed method in identifying the fault is the best, in both groups with/without the added noise $r(t)$.

## 5. Conclusions

The proposed method of fault diagnosis of rolling element bearings named ANFIS-BFDM is presented. The online solution for preprocessing measured data and the way of exploiting the filtered data to label the target domain were our key proposals in this research. In the offline phase, frequency-based splitting of the stream of measured data into different time series was performed to cancel the high-frequency region. The optimal data screening threshold ODST was distilled in remaindered low-frequency data to set up the impulse noise filter FIN. An ANFIS was trained from the preprocessed data to identify the dynamic response of the managed bearing(s). In the online phase, the ODST and ANFIS were employed to filter noise and recognize online the object's health status, respectively. Together with the survey results obtained, some aspects can be observed from the theoretical basis, as follows.

1. This combination of filtering high-frequency noise and IN allows for improving the processing efficiency and speed, suitable for online applications.
2. The proposed way of organizing data in IDS_1 and IDS_2 (12) of the FIN enriches information related to the labeled data samples covering both the source and target domains. It allows not only to weaken the negative influence of the domain drift between the source and target, but also to increase the difference of data correlation between with and without IN to improve the filtering effectiveness.
3. As presented in the proposed algorithm ANFIS-BFDM, the ANFIS that takes the role of the mapping (1) from the fusion domain to the target domain can make the negative impacting degree of domain drift between the source and target domains weaken.

In short, there are two key advantages of the proposed method: (i) the possibility for actual applications of the data preprocessing solution based on SSA and the filter FIN in

online filtering of the measured data, and (ii) the compared effectiveness of the ANFIS-BFDM in reliably identifying a fault even if severe impulse noise appears in the databases. These aspects are verified in Section 4.

Finally, despite the strong points, the considerable time delay related to the calculating cost of this method is also a challenge. The improvement of the delay is the motivation for the authors' future research.

**Author Contributions:** Conceptualization, S.D.N.; methodology, Q.T.T.; software, S.D.N. and Q.T.T.; validation, S.D.N.; formal analysis, Q.T.T.; investigation, S.D.N.; resources, Q.T.T.; data curation, Q.T.T.; writing—original draft preparation, Q.T.T.; writing—review and editing, S.D.N.; visualization, Q.T.T.; supervision, S.D.N.; project administration, S.D.N.; funding acquisition, S.D.N. All authors have read and agreed to the published version of the manuscript.

**Funding:** This research received no external funding.

**Institutional Review Board Statement:** The study was conducted in accordance with the Declaration of Helsinki, and approved by the Institutional Review Board.

**Informed Consent Statement:** Informed consent was obtained from all subjects involved in the study.

**Data Availability Statement:** Data available on request.

**Acknowledgments:** The authors would like to thank the support from the Vietnam National Foundation for Science and Technology Development (NAFOSTED) under grant number 107.01-2019.328.

**Conflicts of Interest:** The authors declare no conflict of interest.

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
