# Peer review of "Bearing Fault Diagnosis Based on Measured Data Online Processing, Domain Fusion, and ANFIS"

_computation, doi:10.3390/computation10090157_

Round 1

Reviewer 1 Report

    The authors presented a very interesting paper on a new online bearing fault diagnosis methodology called ANFIS-BFDM using adaptive neu-fuzzy inference system (ANFIS). In which the reduction of the influence of domain drift between the source and target domains (DDSTD) is taken into account in both data processing and damage identification.The topic deals with the issue of processing on-rail noise in measurement data, from SMD sensors.

The article is needed because a new method is presented. However, the authors should improve minor editorial and substantive comments.

In particular….

1. Introduction

In my opinion, the introduction is a bit long. It would be advisable to divide the introduction into a literature review (showing the current state of research) and proposals proposed by the authors.

However, I think that the bibliography should be gently expanded to include other "important" works in the field of bearing damage diagnostics. Because the issue has not only a typically "algorithmic" dimension, but also a "mechanical" one.

Please check, for example, the works of:

https://doi.org/10.1016/j.ymssp.2019.05.003

G. Yu, "A Concentrated Time–Frequency Analysis Tool for Bearing Fault Diagnosis," in IEEE Transactions on Instrumentation and Measurement, vol. 69, no. 2, pp. 371-381, Feb. 2020, doi: 10.1109/TIM.2019.2901514

S. Zhang, S. Zhang, B. Wang and T. G. Habetler, "Deep Learning Algorithms for Bearing Fault Diagnostics—A Comprehensive Review," in IEEE Access, vol. 8, pp. 29857-29881, 2020, doi: 10.1109/ACCESS.2020.2972859.

nad others…

Not all symbols used in the formulas are explained. Perhaps the authors should introduce nomenclature?

2.1. Building ANFIS from a database

Please clarify exactly what modifications the authors proposed.

Is this case only about Gaussian ker- nel function K(.)?

2.2.1. Related Definitions

Definition 2

Did the authors also check another metric? Why did they use this particular one?

   3.     Proposed Method of DDM-Based Fault Diagnosis

    The authors cite only papers [19] (Nguyen, S.D.; Choi, S.B.; Kim, J.H.   Mechanical Systems and Signal Processing. 2020, 145, Article ID 106958)

    This is logical and understandable. But did the authors also check other works?

Line 285 >>Building datasets for training, testing and checking rely on them. Specifically, by sliding a window (with a width of N data points) along the columns of… << How exactly was the testing and learning data set built in ?. 

4. Evaluating the ANFIS-BFDM

4.1.  Establishing database

I think, in my opinion, the authors should use more cases to assess bearing damage. However, it is sufficient

The structure of writing is improved in accordance with IMRAD-C, namely Introduction, Methods, Results and Discussion, Conclusion. So it's clearer.

Conclusions

The conclusion must answer the purpose. Some recommendations for future work should be given in the Conclusion Perhaps the authors plan to build an integrated decision system for their Research in the future.

Processing online noise in sensor-derived measurement data (SMD) and mitigating the effect of domain drift are always a challenge. 

The authors proposed a new adaptive inference with a positive effect, as shown in an experiment

The results reflect the positive effects of the method, even if there is an increase in impulse noise in the databases.

 The article with minor corrections, in my opinion, should be published, as it is a major scientific contribution in the field of noise processing.

In addition, the paper is mostly based on the work of the author Nguyen [19], who is an established author in this field.

Author Response

Dear Referee #1:

On behalf of the authors, I would like to thank you sincerely. We are very grateful to you for the positive estimation, the useful comments, and the thoughtful suggestions. Based on them, we have modified the original manuscript carefully and hope it has reached your magazine's standard. Once again, we acknowledge your opinions which are valuable in improving the quality of our manuscript.

Sincerely yours,

The corresponding author.

Reviewer 2 Report

The article is devoted to developing a theoretical base within the application-important field of technical diagnostics focused on the online evaluation of the technical condition of rotary machines by determining the value of selected parameters. The article is prepared at a high expert level, the structure is adequate. The article's readability would benefit from adding nomenclature, as the share of abbreviations and symbols is relatively high. As for the list of references, self-citations make up almost a third of all cited works, which is an unusually high ratio, but given the continuity of this article with the previous works of the authors, this is understandable due to continuity.

Author Response

Dear Referee #2:

On behalf of the authors, I would like to thank you sincerely. We are very grateful to you for the positive estimation, the useful comments, and the thoughtful suggestions. Based on them, we have modified the original manuscript carefully and hope it has reached your magazine's standard. Once again, we acknowledge your opinions which are valuable in improving the quality of our manuscript.

Sincerely yours,

The corresponding author.
